# Strong transient magnetic fields induced by THz-driven plasmons in graphene disks

Jeong Woo Han[1], Pavlo Sai [2], Dmytro B. But [2], Ece Uykur [3], Stephan Winnerl [3], Gagan Kumar[4], Matthew L. Chin[5], Rachael L. Myers-Ward[6], Matthew T. Dejarld[6], Kevin M. Daniels[5], Thomas E. Murphy [5], Wojciech Knap[2] & Martin Mittendorff [1] ✉

Strong circularly polarized excitation opens up the possibility to generate and control effective magnetic fields in solid state systems, e.g., via the optical inverse Faraday effect or the phonon inverse Faraday effect. While these effects rely on material properties that can be tailored only to a limited degree, plasmonic resonances can be fully controlled by choosing proper dimensions and carrier concentrations. Plasmon resonances provide new degrees of freedom that can be used to tune or enhance the light-induced magnetic field in engineered metamaterials. Here we employ graphene disks to demonstrate light-induced transient magnetic fields from a plasmonic circular current with extremely high efficiency. The effective magnetic field at the plasmon resonance frequency of the graphene disks (3.5 THz) is evidenced by a strong (~1°) ultrafast Faraday rotation (~20 ps). In accordance with reference measurements and simulations, we estimated the strength of the induced magnetic field to be on the order of 0.7 T under a moderate pump fluence of about 440 nJ cm$^{-2}$.

Light-matter interaction is most commonly described via the electric rather than the magnetic field component of light, even though the magnetic field component of electromagnetic radiation can become significant. In the terahertz (THz) frequency range, metallic metamaterials such as split-ring resonators or spiral structures can enhance the magnetic field component by one to two orders of magnitude[1–4]. The design of the material allows for favoring either strong local enhancement or rather homogeneous magnetic fields over larger areas. Note, however, that these enhanced fields always oscillate with the period of the inducing light field. Generating unipolar short magnetic pulses by optical excitation requires other physical effects such as the inverse Faraday effect[5–11]. In recent years, there have been a number of studies for the demonstration of particularly strong pulses through the interaction between ultra-short laser pulses and phonons[12–19]. To effectively utilize this interaction, the phonon should

not be screened by free carriers such that insulating properties are the inevitable condition as media for the inverse Faraday effect. Thus, most of the reported short magnetic pulses via the inverse Faraday effect have been generated in insulating materials[14–19]. Typically, infrared and THz fluences of the order of mJ cm$^{-2}$ are required to generate effective THz magnetic pulses in the mT range. A recent study on paramagnetic CeCl$_3$ predicts an effective magnetic field in the 100 T range when pumped at a fluence of 40 mJ cm$^{-2}$. Even though this value is orders of magnitude stronger than in earlier studies, the fluence required to generate useful magnetic fields is rather high.

An alternative approach to generate transient magnetic fields is the excitation of circulating currents in conductive media, e.g. plasmas[20]. In this first experiment, several MW of power was dissipated in the medium to generate a plasma and subsequently a magnetic field in the μT range. While more recent measurements on plasmas report

[1]Universität Duisburg-Essen, Fakultät für Physik, 47057 Duisburg, Germany. [2]CENTERA Laboratories, Institute of High Pressure Physics PAS, 01-142 Warsaw, Poland. [3]Helmholtz-Zentrum Dresden-Rossendorf, Dresden 01328, Germany. [4]Indian Institute of Technology, Guwahati, Assam 781039, India. [5]University of Maryland, College Park MD 20740 MD, USA. [6]U.S. Naval Research Laboratory, Washington DC 20375 WA, USA. ✉e-mail: martin.mittendorff@uni-due.de

much higher efficiencies[21], exploiting the magnetic field in the vicinity of the plasma is rather difficult. Instead of a plasma, free charge carriers with linear dispersion relation are promising candidates to enhance the efficiency[22]. Without further patterning, the current induced by the circularly polarized pump radiation is circulating around the area illuminated by the laser beam[23]. Confinement of the circulating currents in plasmonic structures can strongly enhance the efficiency of the inverse Faraday effect as it was shown for gold particles[24] and more recently for plasmonic arrays of gold nanodisks[25] with pump radiation in the visible range. The circularly polarized laser pulse produces the rotational motion of charge carriers, leading to the magnetic pulses[23]. Considering this mechanism, it may pave the way for realizing stronger, optically-driven magnetic pulses via an in-phase motion of charge carriers, which can be realized by the collective motion of free carriers, i.e., plasmon[24,26,27].

Here, we exploit this effect to demonstrate terahertz (THz) optical pulse induced magnetic fields, evidenced by strong and ultrafast Faraday rotation $\theta_F$ in graphene plasmonic disks at a frequency of 3.5 THz, which depends on the dimensions of the disks[28]. In this case the graphene disks have a diameter of 1.2 μm and are fabricated from quasi free-standing bilayer graphene[29,30] on semi-insulating SiC via electron-beam lithography. The dimensions of the disks and the array size were optimized for the experiments at FELBE. The periodicity of 1.5 μm leads to strong light-matter interaction and at the same time limits the coupling between the disks. To generate the transient magnetic fields, we employed circularly polarized pump pulses resonant with the plasmon frequency $\omega_p$. The external electric field of the pump radiation drives circular plasmonic currents in the disks that in turn induce magnetic fields perpendicular to the disks. A Faraday angle $\theta_F$ of more than 1° was achieved at a moderate pump fluence of about 440 nJ cm$^{-2}$ during the

pulse duration of about 20 ps. By combining with the experimental results of the static magnetic field-dependent $\theta_F$ on graphene disks, we estimate a corresponding magnetic field of about 0.7 T.

## Results

### Faraday rotation in static magnetic field

In order to characterize the Faraday rotation in our graphene disks at plasmon resonance, we performed single-beam experiments in static magnetic fields of up to 6 T in Faraday geometry (cf. Fig. 1a). The narrow-band THz pulses are generated by the free-electron laser (FEL) FELBE at the Helmholtz-Zentrum Dresden-Rossendorf, which provides a continuous pulse train that can be tuned to the plasmon frequency, with a repetition rate being 13 MHz. To clarify the role of the circular plasmons for the Faraday rotation, we also measured unpatterned graphene. Note that we only switched the target sample from the graphene disks to the unpatterned graphene without any other change of the experimental condition for direct comparison to the experimental results. Symbols in Fig. 1b show the experimentally measured Faraday rotation in graphene disks and unpatterned graphene as a function of the magnetic field, which was varied from −6 T to +6 T. Interestingly, the Faraday rotation obtained from the disks manifests a distinct profile compared to unpatterned graphene: for low fields the Faraday rotation is proportional to the applied magnetic field while a clear saturation and even a decrease of the Faraday effect is observed for fields above 3 T. In contrast, unpatterned graphene exhibits negative values of $\theta_F$ in positive magnetic fields and the saturable behavior was not observed in the experimental data (though expected for stronger fields).

In order to understand this contrasting behavior between the two cases, we simulated the Faraday rotation for graphene disks and

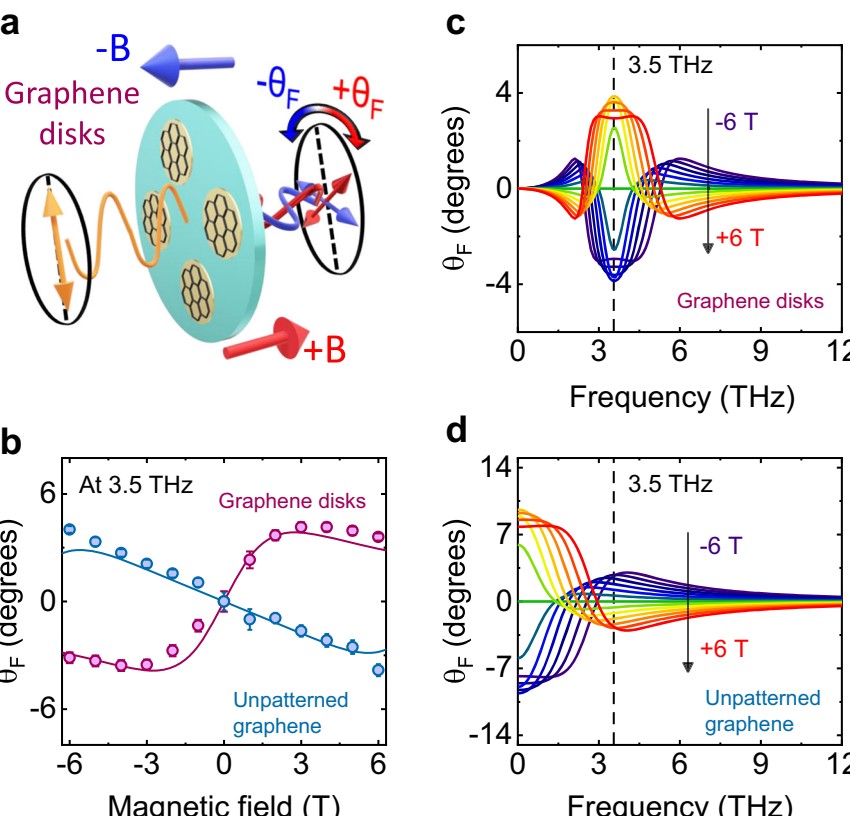

**Fig. 1 | Faraday rotation $\theta_F$ of graphene disks and unpatterned graphene under static magnetic fields. a** Schematic diagram of the experiment for the measurement of $\theta_F$. **b** $\theta_F$ as a function of magnetic fields from −6 T to 6 T. $\theta_F$ is measured at 3.5 THz. Graphene disks and unpatterned graphene are denoted by purple and blue, respectively. Symbols show the experimental data. Lines are simulation results extracted from (**c, d**). Simulation results of $\theta_F$ spectra for graphene disks (**c**), and unpatterned graphene (**d**). Magnetic fields are varied from −6 T to 6 T.

unpatterned graphene, which are provided in Fig. 1c, d, respectively. For the simulation of $\theta_F$ in graphene disks, we derived the optical conductivity tensor $\sigma$ in magnetic fields. The diagonal $\sigma_{xx}$ and off-diagonal components $\sigma_{xy}$ are expressed as:

$$\sigma_{xx} = \frac{fD}{\sqrt{2}} \frac{i\omega(\omega^2 - \omega_p^2 - i\omega\Gamma)}{(\omega^2 - \omega_p^2 + \omega\omega_c + i\omega\Gamma)(\omega^2 - \omega_p^2 - \omega\omega_c + i\omega\Gamma)} \quad (1)$$

$$\sigma_{xy} = \frac{fD}{\sqrt{2}} \frac{\omega^2 \omega_c}{(\omega^2 - \omega_p^2 + \omega\omega_c + i\omega\Gamma)(\omega^2 - \omega_p^2 - \omega\omega_c + i\omega\Gamma)} \quad (2)$$

where $f$, $D$, $\omega_c$, $\Gamma$, $\omega_p$ are filling factor (effective area of graphene), Drude weight, cyclotron frequency, scattering rate, and plasmon frequency respectively. Detailed derivation of Eqs. (1)–(2) and the value of parameters can be referred to in the Supplementary Information. As can be seen in Fig. 1c, the effect for the smallest fields is strongest at resonance, in elevated fields the peak splits into two, leading to the saturation of the Faraday rotation. This splitting is caused by hybridization of the plasmon modes with the cyclotron motion into magneto plasmons. The observed evolution of $\omega_\pm$ is in good agreement with previous studies[31–33]. $\omega_\pm$ can be selectively excited via circularly polarized beams with opposite helicities. In our case, both $\omega_+$ and $\omega_-$ are excited simultaneously since we employed linearly polarized light as the probe beam. From our simulation results on graphene disks, one can observe that the highest $\theta_F$ is achieved nearby $\omega_p$ within the magnetic field range considered in this study.

In contrast, the strongest Faraday rotation is observed in the low frequency range for unpatterned graphene (cf. Fig. 1d). While at lowest frequencies a similar saturation of $\theta_F$ is visible, the Faraday effect quickly decreases for higher frequencies and even changes its sign. The zero crossing is blue-shifted for higher magnetic fields, which is attributed to the increase of the cyclotron frequency $\omega_c$[34,35]. This trend is in accord with the previous reports[36–38]. Note that we employed the semiclassical optical conductivity tensor for the simulation of $\theta_F$ spectra of unpatterned graphene[39,40]. The sign crossover of $\theta_F$ originates from the relative phase between the electric field of the incident THz radiation and the displacement of the charge carriers in the graphene (see Supplementary Information for further explanation). To compare the simulation results to the experimental data, we plot $\theta_F$ as a function of the magnetic field at 3.5 THz as solid lines in Fig. 1b. As can be seen, the simulation reproduces the experimental data well.

## Measurement of pump-induced Faraday rotation
Figure 2 shows a sketch of the experimental setup used to measure the temporal evolution of $\theta_F$. A circularly polarized pump pulse illuminates the sample, while a weaker, linearly polarized pulse is used to probe the pump-induced Faraday rotation. A quarter-wave plate $\lambda/4$ was inserted in the pump beam and was switched from −45° to +45° to produce left ($\sigma^+$) - and right ($\sigma^-$) - handed circularly polarized pump radiation, respectively. In order to quantify the pump-induced Faraday angle $\theta_F$, a wire grid polarizer was positioned behind the sample, allowing the probe beam to separate into two orthogonal components that, in absence of a pump pulse, have equal intensity. Both components were measured by bolometers (B1 and B2), resulting in measurements of the pump-induced change in transmission of each of the two components. A pump-induced Faraday rotation leads to an increase of one component, while the second component is decreased; a pump-induced change in transmission without Faraday rotation would lead to the same change in both detectors.

The main experimental results of the pump-probe experiment are shown in Fig. 3. All measurements were carried out at 10 K. Figure 3a, b show the relative change in transmission as a

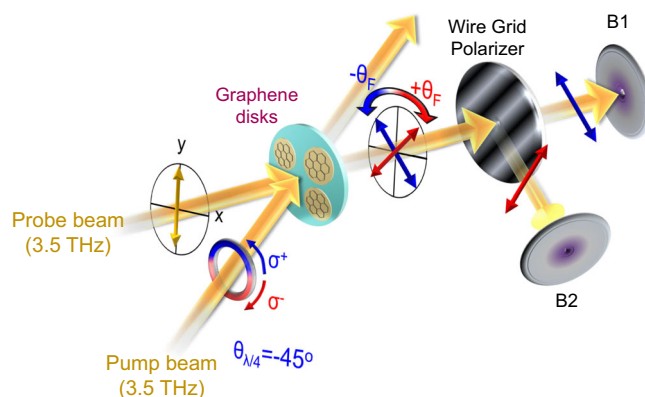

**Fig. 2 | Schematic of the experiment for pump-induced Faraday rotation $\theta_F$ on graphene disks.** The frequency of the probe and pump beam is set to 3.5 THz. A quarter wave plate ($\lambda/4$ plate) is located in the pump beam path. Its rotations of −45° and +45° generate the left ($\sigma^+$) - and right ($\sigma^-$) - handed circularly polarized pump beam. The probe beam is linearly polarized in the vertical direction, the sign of $\theta_F$ denotes its direction. A wire grid polarizer is located in the probe beam path and it is aligned to 45° with respect to the incident probe beam. The reflected and transmitted probe beams from the wire grid polarizer are guided to bolometers B2 and B1, respectively.

function of the delay time, measured at bolometers B1 and B2, respectively, for the left-handed circularly polarized pump beam ($\sigma^+$). The pump fluence was varied in the range of 28 nJ cm$^{-2}$ to 440 nJ cm$^{-2}$. Figure 3d, e show the pump-probe signals measured with right-handed circularly polarized pump beam ($\sigma^-$). For the case of $\sigma^+$ (Fig. 3a, b), the stronger $\Delta T/T$ was observed in the B1 while the case for $\sigma^-$ shows inverted trends (Fig. 3d, e). Note that the summation of $\Delta T/T$'s measured in both geometries should be conserved and corresponds to the regular pump-probe signal, i.e. the pump-induced change of the transmission. We experimentally confirmed that unpatterned graphene gives significantly weaker pump-induced Faraday rotation, emphasizing the role of the plasmonic structure. As discussed above, the difference between subfigures a and b (d and e), is the relevant measure for the pump-induced Faraday rotation, which is calculated in Fig. 3c, f.

As can be seen, the profiles of $\theta_F$ are similar, but their signs are inverted, i.e. the helicity of the plasmonic current and thus the orientation of the magnetic field is well controlled by the polarization of the pump pulse. The different magnitudes observed in the cases of $\sigma^+$ and $\sigma^-$ originate from the imperfection of the circular polarization of the pump beam (see Supplementary Information). The right scale in Fig. 3c, f show the estimated magnetic fields induced by the circularly polarized pump beam, which were inferred from the measurement data in static magnetic fields (cf. Figure 1b).The maximum pump-induced Faraday rotation reached at a fluence of 440 nJ cm$^{-2}$ corresponds to a magnetic field of 0.7 T in the static measurements. While directly measuring the transient magnetic field is beyond the scope of the current study and remains an interesting task for future experiments, our results, together with numerical simulations, suggest that a circularly polarized THz pulse with moderate fluence generates the circular plasmonic current and therewith magnetic fields with a strength on the order of 0.5 T. Measurement at slightly larger disks with lower plasmon frequency revealed smaller pump-probe signals and Faraday rotation, emphasizing the role of the resonance between photon and plasmon frequencies.

## Finite-element simulations of dynamical magnetic field
Unlike the static, homogeneous magnetic field that is produced by superconducting coils, the currents and magnetic fields induced by optical illumination have a spatiotemporal character that is related to

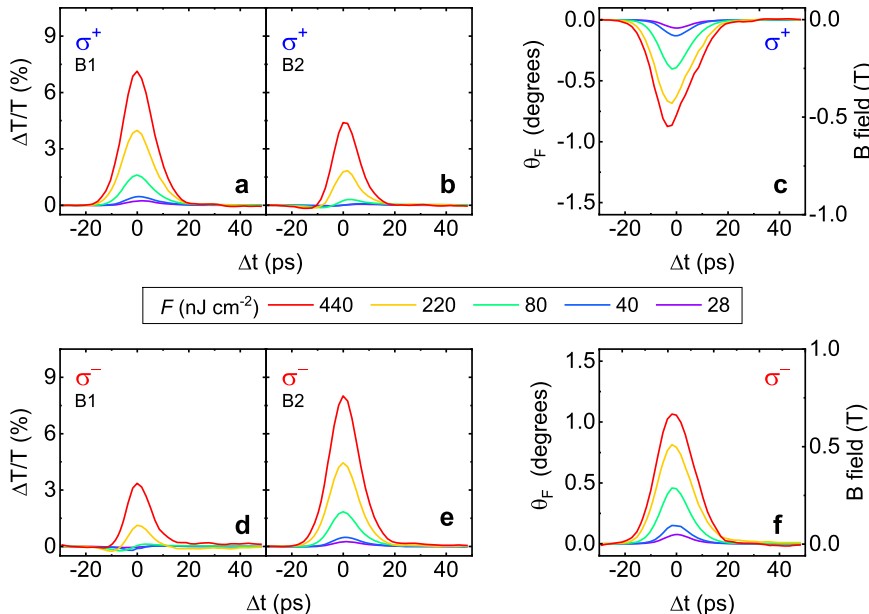

**Fig. 3 | Experimental results of pump-induced transmission change ΔT/T and corresponding θ_F on graphene disks. a**, **b** ΔT/T's measured at the bolometers of B1 and B2. The pump beam is the left-handed circularly polarized (σ⁺). Δt denotes the time delay between the pump and the probe pulses. **d**, **e** Experimental data set corresponding to (**a**, **b**) for the right-handed circularly polarized pump radiation (σ⁻). **c**, **f** θ_F's induced by σ⁺ and σ⁻. The right axes show the corresponding magnetic field. Applied fluences F are denoted in the center of the subfigures.

both the graphene size and the optical periodicity. To further analyze the dynamical magnetic fields, we performed finite element calculations, simulating the spatial and temporal evolution of the magnetic field. In a first step, COMSOL was exploited to calculate the electric field distribution in the vicinity of the graphene disks. The input field was taken to be a monochromatic excitation with intensity equivalent to the highest pump fluence. The parameters for graphene, i.e. the carrier density and mobility, were taken from linear spectroscopy (see Supplementary Information). From the temporally and spatially resolved electric field results, we derived the carrier density distribution within the disks as a function of time (cf. Fig. 4a). From the drift of the carriers around the disk, caused by the plasmonic current, we calculated the three-dimensional magnetic field distribution (cf. Fig. 4b). The magnetic field strength at the position of the graphene disk is depicted by the color code, while the length and the direction of the arrows in Fig. 4b indicate the strength and the direction of the induced magnetic fields, respectively. As mentioned above, carriers are shifted to the edge of the graphene disk and start the rotational motion, which gives rise to the fact that the maximum magnetic field is observed in the area of the strongest currents, i.e. around the charge carriers traveling around the edge, and not perfectly centered. The calculated maximum magnetic field is about 0.35 T, which is on the same order of magnitude as the corresponding experimental value (0.7 T). An animated version of Fig. 4b, showing the temporal evolution of the magnetic field, is available online.

## Discussion

Compared to earlier studies, the generation of the magnetic field observed in our experiment is extremely efficient: a theoretical study predicted magnetic fields in the 1 T range when 2 nm sized plasmonic gold nanoparticles are illuminated with about $5 \cdot 10^{14}$ W m$^{-2}$ [41]. This theoretical prediction was verified in a recent experiment on 100 nm sized colloid plasmonic gold nanoparticles (AuNP) for the observation of optically induced inverse Faraday effect[24]. As shown by Cheng et al., the plasmonic current excited by a 515 nm pulsed laser gives rise to the Faraday rotation θ_F being around 0.1° when excited with a peak intensity of about $10^{14}$ W m$^{-2}$, which corresponds to the magnetic field

of 0.038 T. The Verdet constant of AuNP was confirmed to be about 43 rad T$^{-1}$m$^{-1}$. In contrast to those earlier studies, the graphene-based plasmonic structures are resonant in the THz frequency range. Considering the strength of light-induced magnetic field and peak intensity (~5·10$^8$ W m$^{-2}$) corresponding to the fluence of 440 nJ cm$^{-2}$, the generation efficiency of the magnetic field is about six orders of magnitude higher than ref. 24. From the static magnetic field dependence (Fig. 1), the extracted Verdet constant of graphene disks is about $5 \cdot 10^7$ rad T$^{-1}$m$^{-1}$, which is orders of magnitude higher than the reported value in the strong Faraday rotator terbium doped boron-silicate glasses[42]. Because the spatial scale of optically induced magnetic field distribution strongly depends on the size of the plasmonic medium, we achieved a useful magnetic field that is localized on a μm scale, enabling the potential applications for in-situ experiments. This allows exploiting the magnetic field also for applications beyond Faraday rotation, e.g. by placing molecules or nanoparticles in the vicinity of the disks.

## Methods

### Sample preparation and characterization

Monolayer epitaxial graphene was synthesized by the thermal decomposition, or Si sublimation of semi-insulating 6H-SiC (0.1 deg offcut) at 1580 °C in 100 mbar high purity argon. The reactor is then cooled to 1050 °C, where the SiC is passivated with hydrogen at 900 mbar, decoupling the $6\sqrt{3}$ buffer layer and forming quasi-freestanding bilayer epitaxial graphene, which is p-doped with a carrier density of about $8 \cdot 10^{12}$ cm$^{-2}$ [29]. The graphene was patterned into a 2 mm × 2 mm square array of disks with a periodicity and diameter of 1.5 μm and 1.2 μm, respectively. As the distance between the disks is small compared to the diameter, the plasmonic motion in neighboring disks are coupled, leading to a slightly lower plasmon frequency compared to a single, isolated disk. The patterning was done by electron-beam lithography and a subsequent oxygen plasma etch. For the characterization of graphene disks, we measured the transmission spectrum using Fourier-transform infrared spectroscopy at room temperature and confirmed that the plasmon frequency is located in nearby 3.5 THz. Transmission data and fitting results can be found in the Supplementary Information. Next to the disk array, a section of the

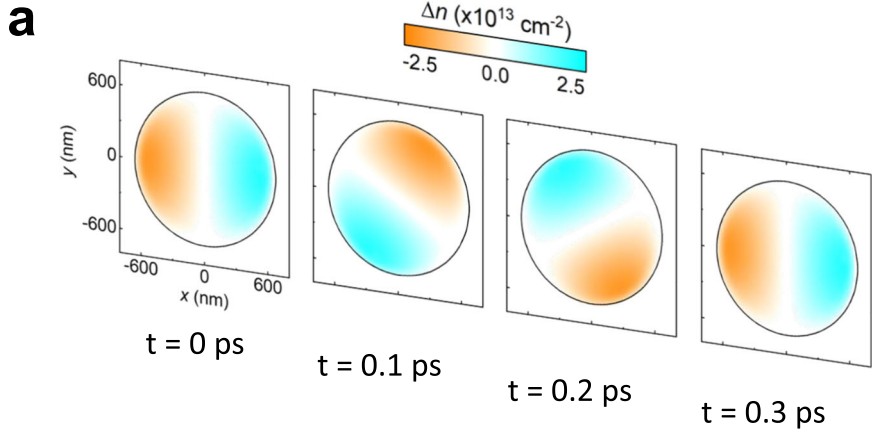

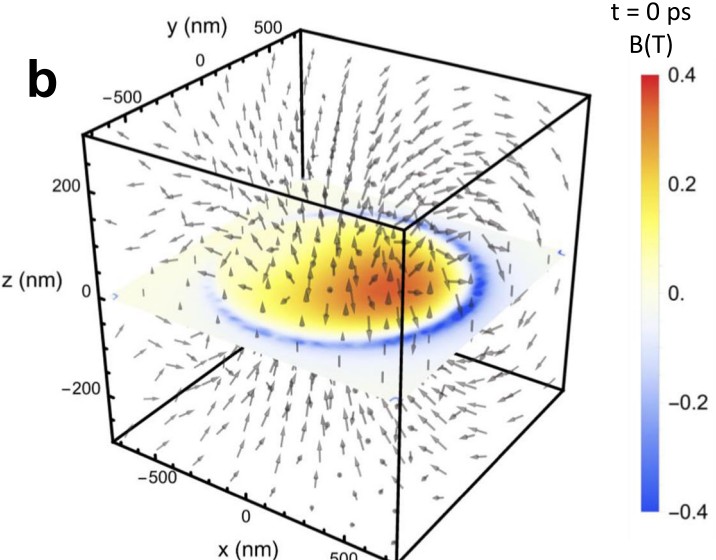

**Fig. 4 | Simulation result of carrier distribution and magnetic field induced by circularly polarized pump beam on graphene disk. a** Time t dependent carrier density Δ*n*. The frequency of pump and probe beams employed in the simulation was set to the equivalent value of our experimental condition (3.5 THz). **b** Three-dimensional magnetic field B distribution near graphene disk at *t* = 0. The strength of the magnetic field of the surface of graphene disk is illustrated by color code.

sample was left unpatterned, i.e. large area graphene of the exact same type, in order to enable quantitative comparison between patterned and unpatterned graphene.

The Faraday angle measurements in static fields were carried out in an optical magneto cryostat for magnetic fields of up to 6 T. The sample was placed in a low-pressure Helium atmosphere to keep the temperature constant at 10 K. To measure the Faraday rotation in static magnetic fields, the transmission through a linear polarizer behind the sample was measured as a function of the polarizer angle. The free-electron laser (FEL) at the Helmholtz-Zentrum Dresden-Rossendorf served as radiation source for the static and pump-probe experiments. It provides a continuous pulse train of narrow band THz light with a repetition rate of 13 MHz and a pulse duration of about 9 ps (FWHM).

We measured the helicity of the employed pump beam, which turns out to have a linear component of about 15%. Hence, our pump beam is slightly elliptical, the contribution of the linear component is negligible for the observation of the pump-induced Faraday rotation (see Supplementary Information for details). Note that the FEL frequency was slightly adjusted for best circular polarization. All measurements are performed at 10 K.

## Magnetic field calculation

The electromagnetic fields were calculated in COMSOL using a finite element method solving Maxwell's equations[43]. The simulation used periodic boundary conditions in the horizontal direction, to simulate the square lattice of graphene elements, and an absorbing boundary in the silicon carbide to mimic a semi-infinite substrate. Specifically, we used the RF module to calculate the z-component of the electric field 50 nm (ξ) above the graphene disk, considering periodic boundary conditions. Note that we performed the simulation at resonance, i.e. $\omega_{simul}$ = 2π 3.5 THz, where the imaginary part of the complex conductivity vanishes and in-plane fields are partially screened. Thus, the in-plane components ($E_x$ and $E_y$) of electric field distribution within the disk can be neglected. The value of the incident plane wave $\mathbf{E}_{THz}(t)$ was set to a value of $E^0_{THz}$ = 5.75 · 10$^5$ V m$^{-1}$, which is equivalent to the highest field strength in the pump-probe experiments:

$$\mathbf{E}_{THz}(t) = E^0_{THz}(\sin(\omega_{simul}t)\hat{\mathbf{E}}_x \pm \cos(\omega_{simul}t)\hat{\mathbf{E}}_y) \tag{3}$$

where the sign of the cosine term determines the rotation direction of the polarization. To ensure accurate calculations, we accounted for the

finite thickness of the graphene disk by treating it as a 3D structure with a thickness of $\eta = 10$ nm. Considering the value of $\eta$, the 3D optical conductivity can be converted into 2D ($\sigma_{3D} = \sigma_{2D}/\eta$)[44,45].

To derive the carrier density from the electric field distribution, we considered small changes of the carrier concentration on the scale of the mesh size (16 nm²) of our calculation. This simplifies the relation between the z-component of the electric field and the charge Q per mesh to

$$\frac{E_z(x,y)}{\alpha} = \frac{1}{4\pi\varepsilon_0} \cdot \frac{Q(x,y)}{\xi^2} \qquad (4)$$

where $\varepsilon_0$ is vacuum permittivity. The pump-induced difference in carrier density ($\Delta n$) is calculated by dividing the extracted $Q$ with the mesh area employed in the simulation. The additional factor $\alpha = 1.5 \cdot 10^4$ accounts for the ratio ($\alpha = E_I/E_O$) between the electric field $E_I$ generated from homogeneously distributed carriers and the field $E_O$ caused by the charges within a single mesh: $E_I$ is in this case the electric field calculated via the finite element simulation. To derive the electric field caused by the charges within a single mesh directly underneath the electric field, we calculate the ratio $\alpha$ between the electric field caused by the charges in a single mesh and a homogeneous charge distribution with the same carrier density. By assuming that the charge carrier density is rather constant on the length scale of the mesh size, we can convert the electric field $E_I$ into the field $E_O$, which represents the electric field caused by the charges in the mesh. The current density $\vec{J}$ is given by:

$$\vec{J}(x,y) = (\Delta n(x,y) + \Delta n_{min}) \cdot 2\pi \sqrt{x^2 + y^2} \cdot \omega_{simul} \widehat{r_\perp} \qquad (5)$$

To convert from the pump-induced change in carrier concentration to the absolute number of moving carriers, the minimum value $\Delta n_{min}$ is added to $\Delta n$. In a steady state solution, all charge carriers oscillate in the same circular motion with the angular frequency of the plasmon. In our calculation, we used Cartesian coordinate and $\widehat{r_\perp}$ is the unit vector of the direction of current density represented in Cartesian coordinate. Note that the current density caused by the circular plasmons curls around the center of the disk. Hence, the unit vector of $\widehat{r_\perp} = \frac{(\pm y, \mp x)}{\sqrt{x^2 + y^2}}$ points into the azimuthal direction with the upper sign for clockwise rotation and the lower sign for counter-clockwise rotation. For calculating the 3D distribution of magnetic fields, we employed the Biot-Savart law.

## Data availability

The datasets for this study are available from the corresponding author on request.

## Code availability

The source code for the data analysis, e.g. for the calculation of the magnetic fields, is available from the corresponding author on request.

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

## Acknowledgements

This study was funded by the Deutsche Forschungsgemeinschaft (DFG, German Research Foundation)—Project-ID 278162697—SFB1242. This work was also partially supported by CENTERA Laboratories in the frame of the International Research Agendas Program for the Foundation for Polish Sciences co-financed by the European Union under the European Regional Development Fund (no. MAB/2018/9). Work at the US Naval Research Laboratory was supported by the Office of Naval Research. We thank J. Michael Klopf and the ELBE team for their assistance. Furthermore, we acknowledge technical support by Christoph Böttger.

## Author contributions

M.M. and W.K. conceived and supervised the project. S.W. and M.M. designed the experiment. J.W.H., P.S., D.B.B. and M.M. performed the experiments in static fields as well as the pump-probe measurements. J.W.H. and M.M. interpreted the results with contributions from all co-authors. P.S. and D.M. carried out the COMSOL simulation. G.K. performed the CST simulation for designing the graphene disks. E.U. carried out the FT-IR characterization. R.L.M.-W., M.T.D., and K.M.D. grew the bilayer graphene and M.L.C. fabricated the graphene disks. J.W.H., S.W., and M.M. wrote the manuscript. All co-authors discussed the results and commented on the manuscript.

## Funding

## Competing interests

The authors declare no competing interests.
