## [Peer Review File · Nature Communications]

Reviewers' Comments:

Reviewer #1:

Remarks to the Author:

The paper is interesting and nicely extends recent explorations into the inverse Faraday effect (IFE) in nanoscale optical system to the THz regime. On the whole, I think the authors need to be very careful about emphasizing the difference between the "effective" magnetic fields and the "actual" optically induced magnetic field. This is a subtle and confusing point, especially for novice readers, that is worth taking special care to clarify.

For example, on pg 08, the authors are incorrect to state that "Our results demonstrate that sub-Tesla magnetic fields can be generated with a moderate pump fluence." The authors have not measured any magnetic fields in their experiments. They have determined that the graphene has been magnetized to the same amount that it would be if a 0.7 T magnetic field had magnetized the graphene. It is unclear what strength of magnetic field was actually produced by the circularly polarized optical pulse, because graphene does not generate a magnetic field of 0.7 T when it is magnetized by a magnetic field of 0.7 T. To use an electrostatic analogy, the polarization induced in a dielectric medium under an applied 7V potential (e.g., in a capacitor) does not entail that the dielectric medium generates a 7V potential if it is somehow polarized to the same amount in the absence of an applied electrostatic field.

The authors seem to be assuming that there is an electric current generated by the optical pulse that produces a magnetic field of 0.7 T that then magnetizes the graphene sample. This assumption should be stated. Moreover, the discrepancy between finite element simulations and the measured value from the pump-probe IFE experiment here, and the discrepancies between other calculations and IFE measurements reported to date, suggest direct measurements of the optically-induced magnetic field are required to confirm the strength of magnetic field that is actually produced by the optical pulse.

I think this manuscript is worthy of publication, but I do think the authors need to state explicitly that the experiment cannot determine what strength of magnetic field was produced by the optical pulse, and they should better emphasize the explicit relationship between the measured Faraday rotation and the induced magnetization (not necessarily the actual magnetic field). I have listed a few other comments that may aid clarity.

1. All measurements were done at 10 K. This could be important information to mention before the methods section. The authors show in the SI that the signal is smaller at room temperature.
2. Can the authors report any relevant sample characterization? For example, how was the plasma frequency of the graphene determined?
3. (Fig 3) I wonder if the authors can make claims about the lifetime of the IFE signal. It seems that this is not possible. One optical period is ~ 0.3 ps, but the optical pulse is tens of ps long.
4. (Fig 3) Is the different magnitude of θ_F / B field in Fig 3c, f a result of measurement imperfection, or something real? The authors mentioned the CP beam produced was not pure, Fig S4.
5. (Fig 4) Did the authors include the linearly polarized probe beam in their simulations. I might guess that the probe beam has no effect on the results? If not, Fig 4a is confusing. In fact, I believe they used a CW incident field in their simulations. What they called "delta_t" in Fig

4a seems to have nothing to do with the "delta_t" in the experiment (Fig 3).

6. (Line 217, Magnetic field calculations in methods section) Why is E_z relevant in here instead of E_x and E_y ? I do not fully understand the model.

7. pg 7 - "Both components were measured by bolometers (B1 and B2), resulting in measurements of the ΔT pump-induced change in transmission of each of the two components. A pump-induced Faraday rotation leads to an increase of one component, while the second component is decreased; a ΔT pump-induced change in transmission without Faraday rotation would lead to the same change ΔT in both detectors." Is this sufficient to disentangle all other non-linear effects? In reference 24, the authors analyzed the phase relationship between

pump and probe beams to isolate the contribution from the optical Kerr effect in the signal.

Reviewer #2:

Remarks to the Author:

The manuscript presents experimental demonstration of light induced transient magnetism due to Inverse Faraday Effect (IFE) in graphene disks with hundreds of nm radius deposited on SiC as captured through Faraday rotations of a weak probe pulse while excited at plasmon frequency using circularly polarized pump pulses of fluences varying from tens to hundreds of nJ per cm square. The experimental results are supported by electromagnetics simulations using finite element method (FEM).

While using pump-probe technique to capture IFE is not new, the novelty of this work is the application of this technique to arrayed graphene disks which, based on the data presented in the manuscript, the authors performed successfully. Hence, the experimental results are of significant importance.

In my humble opinion, there are following points which should be addressed before considering publication of the manuscript.

[1] The graphene disks have a diameter of $1.2 \mu\text{m}$ and they are separated $1.5 \mu\text{m}$ away from their neighbors. (a) Why did the authors chose this values for the fabrication of the experimental samples? This allows the circular plasmonic current generated in one disk (when excited by the circularly polarized pump pulse) to couple with the same in its neighbor. (b) Is this coupling crucial for the generation induced magnetization?

[2] Whether the generation of the magnetic field is due to the plasmonic current created in each of the graphene discs or due to a current resulting from the coupling of the neighboring disks (and hence circulating around a domain bigger than the area of a single isolated disk) is not clear! Can the authors comment on this?

[3] Is the generation of the magnetic field a plasmonic phenomena? Did the authors perform measurements with non-plasmonic frequency of the pump laser? How difficult is this to achieve in FEL? To ensure that the highest magnetic field is obtained at resonance it would have been instructive to have a comparison with the non-resonant case.

[4] There are several points concerning the FEM simulations which should be addressed in order to support the fact that calculation gave a magnetic field of the same order of magnitude as in the experiments.

(a) Why the electric field is calculated 50 nm above the graphene disk?

(b) Why the in-plane component E_x and E_y are neglected? Isn't they circularly polarized excitation is polarized along X-Y plane? It is the E_x and E_y component which should give rise to the in-plane plasmonic current circulating on the graphene sheet giving rise to a magnetization along the out-of-plane i.e., +/- z direction! What is the mathematical expression of the incident circularly polarized field that is exciting the graphene disk? Also in which plane is the graphene disk situated?

(c) Do the theoretical simulations have the same periodicity ($1.5 \mu\text{m}$) as in the experiments? It is certainly not the case if the bounding box presented in figure 4 do represent the unit cell. Then why the periodicity is chosen to be different from the same in the experiment?

(d) Is the periodic boundary condition in three dimension (3D) or is it a two-dimensional periodicity? If it is in 3D then did the authors check the convergence of the results with respect to the vacuum along the z direction in the simulation cell? If not, they should. The length of the arrows indicating the strength of the induced magnetic field do not show significant decrease while going away from $z=0$ plane suggesting that neighboring cells along z-directions has strong influence on the generated magnetic field.

(e) What does the unit cell (16 nm square) correspond to? The bounding box shown in the figure 4 has a dimension of approximately $600 \text{ nm} \times 1200 \text{ nm} \times 1200 \text{ nm}$.

(f) It is not clear what do E_1 and E_0 correspond to! Please provide mathematical expressions to define them properly, or explain better.

- (g) Is the current density \vec{J} is along the Cartesian y -direction as suggested by \hat{j} ? This would imply that the current is curl free and can not give rise to a magnetic field!
- (h) Is conversion of the optical conductivity from 3D to 2D a standard one? If so, please give proper reference to support it, otherwise please give further justification.
- (i) Please provide details of the equations that are solved and of the numerical methods employed in the FEM code, at least in the supplementary material.

Reviewer #3:

Remarks to the Author:

I believe that the work contains sufficient novelty and broad impact to warrant publication in a journal such as nature communications.

There are, however, several issues which I feel need to be addressed before I can give my recommendation for publication. Please see the attached PDF document for my comments to address.

If the authors are able to address these concerns, then the manuscript would obtain my support in recommendation for publication.

Han *et al* present the use of a graphene microdisk array for the generation of high-intensity magnetic field pulses via plasmon-resonance enhanced inverted faraday effect without the need for the high pump intensities of previous reports. They deduce the field intensity produced via the faraday rotation of a probe beam and reveal the field-dynamics via polarization-resolved variable delay pump-probe spectroscopy. Further the work presents the advantageous of the microfabricated graphene arrays when compared to un-patterned graphene and accomplishes this using industry standard techniques (EBL, RIE) and substrates (SiC) which offer the potential for expansion of these concepts into many applications.

With this noted, I believe that the work contains the novelty and broad impact to warrant publication in a journal such as nature communications.

There are, however, several issues which should be addressed before I can give my recommendation for publication.

1. Literature support needs to be improved in order to appeal to a broad audience. For instance
2. Continuing from the previous point, the authors undersell the applicability of their arrays. They mention, “by placing molecules or nanoparticles in the vicinity of the disks” but do not elaborate on (for instance) the potential for *in-situ* control allowed by optically induced magnetization.
3. There is a significant overlap between Fig. 1a and Fig. 2. In truth it appears as though Fig. 1a is a subset of Fig. 2 with the pump-beam removed and does not add any extra information to the manuscript, and as the bolometers are used for the experiment presented in Fig. 1b I question if Fig. 1a’s inclusion is needed. And to a lesser extent the Fig. 4 insert, especially as it is presented after the transient data of Fig. 3.
4. Can the authors comment on how the error bars in Fig. 1b were determined?
5. Can the authors comment on the relation between ω_p and the dimensions? In section S.1 the authors determine ω_p via FTIR spectroscopy, is there theoretical basis for the shift compared to unpatterned graphene?
6. Fig. 1c&d present the calculated faraday rotation spectra for patterned/unpatterned graphene respectively. There is, however, no experimental data to confirm this. Is this because it is too experimentally tedious (*e.g.* would require a major reconfiguration of the FEL)?
7. Regarding the last two points, were the array dimensions optimized to work with the FEL? If so, the authors should make this clear.
8. Why was 440 nJ cm^{-2} the upper limit of pump fluence? Is it a limitation of the experimental setup?
9. How exactly is θ_f determined (from $\frac{\Delta T}{T}$)?
10. Following from the previous point. In Fig.3 (& several supp figures) $\frac{\Delta T}{T}$ and θ_f are not symmetrical for σ_{\pm} . Can the authors elaborate on this point? Pump ellipticity is discussed (~15%) however, the authors declare this effect to be negligible in the results. The authors should be careful to consider all other potential aspects of the experiment (*e.g.* Transmission dichroism, B1/B2 responsivities, disk fabrication astigmatism)

11. Following on from the previous point, would it be possible to tailor the ω_p based upon disk dimensions. How would fabrication astigmatism affect this (and the resultant inverted faraday-effect)?
12. The material & device preparation methodology seems very similar to the author's previous work by Woo *et al.* (Advanced Photonics Research 3.2 (2022): 2100218), This is peculiarly not referred to in the manuscript.
13. In section S6 polarization nonlinearity contribution to the signal is measured with a linearly polarized pump. The authors then use this as evidence that the contributions of the linear component of the pump to be negligible. This assumption hinges on another assumption that the circular polarized component also has negligible contribution to the plasmonic nonlinearity. This assumption is not supported by the experiment. A more complete picture would be achieved by including circular dichroism and/or faraday ellipticity to account for these contributions.
14. The Authors present experimental (and theoretical) results for Faraday rotation unpatterned graphene, however any results for the inverted faraday effect (*i.e.* Fig. 3) for unpatterned graphene are omitted. I believe this is because the off-diagonal optical conductivity (σ_{xy}) for unpatterned graphene around 3.5 THz to be negligible small. There is however, no rationale present in the text for its omission.
15. What is the significance of the green filled trendline in figure 4? (Under the transient-snapshots)
16. Fig. S5 shows the temperature dependent induced faraday rotation, which the Authors use to determine the induced Magnetic field. Was the (simple) faraday rotation measured at each temperature prior to this measurement? If so, then this should be stated more clearly. If not, the Authors need to support why they expect the rotation measure at 10K is unchanged at elevated temperatures.
17. The authors should consider adding some form of microscopy of the arrays (either optical or SEM) to both enhance the reader understand the structure and to help qualitative evaluation of the structures.

If the authors are able to address these concerns, then the manuscript would obtain my full support in recommendation for publication.

Response to the reviewers

We are grateful for the reviewers careful reading of the manuscript and the helpful comments provided. Indeed, we believe that the points raised by the reviewers are relevant and believe that the revised version of the manuscript is significantly improved compared to the initial submission. In the following we address (black) the reviewers' comments and questions (blue). Changes in the manuscript are highlighted in yellow.

Referee 1

Introductory Comment:

The paper is interesting and nicely extends recent explorations into the inverse Faraday effect (IFE) in nanoscale optical system to the THz regime. On the whole, I think the authors need to be very careful about emphasizing the difference between the "effective" magnetic fields and the "actual" optically induced magnetic field. This is a subtle and confusing point, especially for novice readers, that is worth taking special care to clarify. For example, on pg 08, the authors are incorrect to state that "Our results demonstrate that sub-Tesla magnetic fields can be generated with a moderate pump fluence." The authors have not measured any magnetic fields in their experiments. They have determined that the graphene has been magnetized to the same amount that it would be if a 0.7 T magnetic field had magnetized the graphene. It is unclear what strength of magnetic field was actually produced by the circularly polarized optical pulse, because graphene does not generate a magnetic field of 0.7 T when it is magnetized by a magnetic field of 0.7 T. To use an electrostatic analogy, the polarization induced in a dielectric medium under an applied 7V potential (e.g., in a capacitor) does not entail that the dielectric medium generates a 7V potential if it is somehow polarized to the same amount in the absence of an applied electrostatic field. The authors seem to be assuming that there is an electric current generated by the optical pulse that produces a magnetic field of 0.7 T that then magnetizes the graphene sample. This assumption should be stated. Moreover, the discrepancy between finite element simulations and the measured value from the pump-probe IFE experiment here, and the discrepancies between other calculations and IFE measurements reported to date, suggest direct measurements of the optically-induced magnetic field are required to confirm the strength of magnetic field that is actually produced by the optical pulse. I think this manuscript is worthy of publication, but I do think the authors need to state explicitly that the experiment cannot determine what strength of magnetic field was produced by the optical pulse, and they should better emphasize the explicit relationship between the measured Faraday rotation and the induced magnetization (not necessarily the actual magnetic field). I have listed a few other comments that may aid clarity.

We thank the referee for acknowledging the importance of our work. Moreover, we appreciate his/her helpful comments that have made our manuscript better. We note that graphene, similar to graphite, features a strong diamagnetic response [doi.org/10.1103/PhysRevLett.105.207205, doi.org/10.1103/PhysRevB.91.094429] and is capable for magnetic levitation [doi.org/10.1088/1361-6463/ac683c]. The magnetic susceptibility in graphene depends on temperature, magnetic field and impurity concentration and can reach values of up to $-4 \times 10^{-7} \text{ m}^3/\text{kg}$ (based on mass density) [doi.org/10.1103/PhysRevB.91.094429], corresponding to dimensionless values $\chi \approx -9 \times 10^{-4}$. Consequently, the induced magnetization is negligible and we assume in the following that the magnetic field in the sample is similar to the applied external field. Regardless of the magnetization, as the reviewer pointed out, 0.7 T is not a measured value, such that the statement on page 8 (Our results demonstrate that....) might be misleading. We revised this in the following way to include our assumption and the future task based on the reviewer's comment:

“The right scale in Fig. 3c and f show the **estimated** magnetic fields induced by the circularly polarized pump beam, which were **inferred** from the measurement data in static magnetic fields (cf. Fig. 1b).”

“**While directly measuring the transient magnetic field is beyond the scope of the current study and remains an interesting task for future experiments, our results, together with numerical simulations, suggest that a circularly polarized THz pulse with moderate fluence generates the circular plasmonic current and therewith magnetic fields with a strength on the order of 0.5 T.**”

1. All measurements were done at 10 K. This could be important information to mention before the methods section. The authors show in the SI that the signal is smaller at room temperature.

We agree that this is vital information that should not only show up in the methods section and added this information on page 7:

“... are shown in Fig. 3. All measurements were carried out at 10 K. Figure 3a.....”

2. Can the authors report any relevant sample characterization? For example, how was the plasma frequency of the graphene determined?

For the characterization of the graphene disks, we measured the transmission spectrum at room temperature using FTIR (see Fig. S1) and carried out the fitting based on the plasmon model for graphene (Eq. 1). From this, one can extract the three parameters relevant for modeling, i.e., Drude weight D , scattering rate Γ , and plasmon frequency ω_p . Assuming a Drude-like conductivity from those parameters one can extract the Drude weight and the carrier density (cf. table S1 in the SI). We would like to point out that the same type of sample was characterized in the past by additional means, e.g. Raman spectroscopy and atomic-force microscopy (cf. J. Phys. Photonics 3 (2021) 01LT01).

3. (Fig 3) I wonder if the authors can make claims about the lifetime of the IFE signal. It seems that this is not possible. One optical period is ~ 0.3 ps, but the optical pulse is tens of ps long.

The lifetime of the IFE is directly linked to the plasmon lifetime, which is significantly shorter than the FEL pulse duration. The spectral width of the plasmon absorption is measured to be around 1.48 THz (FWHM), from which we can derive the plasmon lifetime assuming Gaussian shape via the bandwidth-lifetime product to be 0.3 ps (FWHM), which is very close to the optical period (0.29 ps). Thus, the temporal resolution of the experiment is far too low to actually measure the plasmon lifetime.

4. (Fig 3) Is the different magnitude of θ_F / B field in Fig 3c, f a result of measurement imperfection, or something real? The authors mentioned the CP beam produced was not pure, Fig S4.

Indeed, we interpreted the difference magnitude in Figs. 3c and f to be caused by experimental imperfections, mostly the polarization of the pump beam. To clarify this point to the readers, we added the following statement on page 8 (140):

“...the pump pulse. The different magnitudes observed in the cases of σ^+ and σ^- originate from the imperfection of the circular polarization of the pump beam (see supplementary information). The right scales...”

5. (Fig 4) Did the authors include the linearly polarized probe beam in their simulations. I might guess that the probe beam has no effect on the results)? If not, Fig 4a is confusing. In fact, I believe they used a CW incident field in their simulations. What they called " Δt " in Fig4 a seems to have nothing to do with the " Δt " in the experiment (Fig 3).

Our simulation aimed to capture the carrier motion induced by the circularly polarized pulse, thus we did not include a probe pulse (or how it would be influenced) in the simulation. From the circular currents, we estimated the spatial distribution of the magnetic fields generated by the circular plasmonic current. We completely agree with the reviewer that in this aspect Δt is not a proper notation. As the inset of Fig. 4 gives rise to confusion, we omitted it in the revised version.

6. (Line 217, Magnetic field calculations in methods section) Why is E_z relevant in here instead of E_x and E_y ? I do not fully understand the model.

Since we simulated at 3.5 THz (at resonance with the plasmon frequency), the imaginary part of optical conductivity is zero. This suggests that the carriers oscillate with the incident circularly polarized THz beam without phase retardation, leading to the perfect screening effect on the in-plane components of electric field. We agreed that the explanation for this effect is unclear and added the explanation on page 13 (221-222) as:

“Note that we performed the simulation at resonance, i.e. $\omega_{\text{simul}} = 3.5$ THz, where the imaginary part of the complex conductivity vanishes and in-plane fields are fully screened. Thus the in-plane components (E_x and E_y) of electric field distribution within the disk can be neglected.

7. pg 7 - "Both components were measured by bolometers (B1 and B2), resulting in measurements of the pump-induced change in transmission of each of the two components. A pump-induced Faraday rotation leads to an increase of one component, while the second component is decreased; a pump-induced change in transmission without Faraday rotation would lead to the same change in both detectors." Is this sufficient to disentangle all other non-linear effects? In reference 24, the authors analyzed the phase relationship between pump and probe beams to isolate the contribution from the optical Kerr effect in the signal.

Indeed, the difference in the signal of the bolometers alone is not sufficient by itself as other nonlinear effects can lead to a change in polarization, e.g. a residual linear component in the pump radiation. To make sure that these are not dominating the measured signal, we employed linear pump-probe measurements (as shown in the SI) in section S6. This would also include any anisotropic heat distribution in the charge carriers that might be present in the main measurements. Beyond these points, there are to the best of our knowledge no additional effects that might lead to a pump-induced variation of the polarization.

Response to the comment of the referee 2

Introductory Comment:

The manuscript presents experimental demonstration of light induced transient magnetism due to Inverse Faraday Effect (IFE) in graphene disks with hundreds of nm radius deposited on SiC as captured through Faraday rotations of a weak probe pulse while excited at plasmon frequency using circularly polarized pump pulses of fluences varying from tens to hundreds of nJ per cm square. The experimental results are supported by electromagnetics simulations using finite element method (FEM).

While using pump-probe technique to capture IFE is not new, the novelty of this work is the application of this technique to arrayed graphene disks which, based on the data presented in the manuscript, the authors performed successfully. Hence, the experimental results are of significant importance.

In my humble opinion, there are following points which should be addressed before considering publication of the manuscript.

We thank the referee for evaluating our work as the interesting study not only for scientific point of views but also for application perspectives.

1. The graphene disks have a diameter of $1.2\ \mu\text{m}$ and they are separated $1.5\ \mu\text{m}$ away from their neighbors. (a) Why did the authors chose this values for the fabrication of the experimental samples? This allows the circular plasmonic current generated in one disk (when excited by the circularly polarized pump pulse) to couple with the same in its neighbor. (b) Is this coupling crucial for the generation induced magnetization?

First of all, the underlying effect does not require an array of disks, a single disk would also show the same behavior as the array. The array is only necessary for the experiment as the spot size is about $1\ \text{mm}$ and a single disk would not be suitable for this kind of measurements. Thus, an array of disks was fabricated in order to reach a light matter interaction suitable for this experiment. The denser the disks are packed, the higher is the coverage and the measurable signatures. At the same time, this leads to coupling between the disks as pointed out by the reviewer, which leads to a red shift of the plasmon frequency. To remain in a regime that enables strong light-matter interaction with only moderate coupling of the disks, we chose the parameters of the periodicity of $1.5\ \mu\text{m}$. To represent the role of the coupling for the resonance frequency and the light matter interaction, we performed CST simulations as shown in Fig. R1. The chosen spacing of $1.5\ \mu\text{m}$ causes a small red-shift of the plasmon resonance, but does not otherwise alter the physics of the interaction. As the coupling is not crucial for the observation of circular plasmons and goes beyond the scope of the current study, we have not added a detailed discussion in the manuscript.

Fig. R1: (a) Resonance frequency of an array with graphene disks with a radius of 600nm in dependence of the periodicity. (b) Transmission as the function of the frequency for various periodicity.

2. Whether the generation of the magnetic field is due to the plasmonic current created in each of the graphene discs or due to a current resulting from the coupling of the neighboring disks (and hence circulating around a domain bigger than the area of a single isolated disk) is not clear! Can the authors comment on this?

Indeed, from the experiment alone we cannot conclude the role of the coupling between the disks. But as mentioned above, in general the plasmonic currents do not require an array or coupling of disks. As can be seen in the simulations in the main manuscript (Fig. 4 (a) and (b)), the electric and magnetic fields are strongly confined to the disk itself. While there is an interaction between neighboring disks (as also shown above), this effect is not crucial for the underlying effect of the circulating plasmonic currents. We would like to further point out, that the same physics was observed in Ref. 24 (<https://doi.org/10.1038/s41566-020-0603-3>) in gold nano spheres that can be considered to be entirely uncoupled due to the low concentration in the solution.

3. Is the generation of the magnetic field a plasmonic phenomena? Did the authors perform measurements

Fig. R2: Experimental results of pump-induced transmission change $\Delta T/T$ and corresponding θ_F on graphene disks with the radius of 620 nm.

with non-plasmonic frequency of the pump laser? How difficult is this to achieve in FEL? To ensure that the highest magnetic field is obtained at resonance it would have been instructive to have a comparison with the non-resonant case.

This is a very good point as we demonstrated the frequency dependence of the plasmonic response already in earlier measurements. The FEL itself is not limiting this type of measurements, but the necessary retardation plates are: the quartz plates are designed for a specific wavelength and the best circular polarization is achieved only in a very narrow spectral range. This is also the reason for the imperfection of the circular polarization discussed above: tuning the FEL to match exactly the design wavelength of the quartz plate was not possible due to water absorption lines. In order to still perform a control experiment, we performed the same measurements with slightly larger disks (1240 nm instead of 1200 nm), and thus a slightly lower plasmon frequency, keeping the FEL tuned to 3.5 THz, i.e. slightly above the resonance. As

can be seen in Fig. R2, the signals are smaller than the ones achieved with the disks at resonance, matching the expectations from theoretical calculations (cf. Fig. 1(c) in the main manuscript). We added the following statement on page 8:

“... generated with a moderate pump fluence. Measurement at slightly larger disks with lower plasmon frequency revealed smaller pump-probe signals and Faraday rotation, emphasizing the role of the resonance between photon and plasmon frequencies.”

(a) Why the electric field is calculated 50 nm above the graphene disk?

Our simulation aimed for calculating the magnetic field distribution within the graphene disks. For this purpose, we simulated the graphene as a thin conductive layer with a finite thickness. In the center of that layer, the electric field would actually be zero, only above the graphene layer we can get a reliable measure of the electric field. In order to avoid artifact stemming from the charge distribution within the disk, we decided to evaluate the field in a distance of 50nm, which is a small distance on the scale of the diameter of the disk (1200nm). The distance to the graphene layer was taken into account in Eq. (3). When the distance to the disks is further increased, the field gets more washed out as charge distribution in the disk is not homogeneous on a relevant length scale anymore.

(b) Why the in-plane component E_x and E_y are neglected? Isn't they circularly polarized excitation is polarized along X-Y plane? It is the E_x and E_y component which should give rise to the in-plane plasmonic current circulating on the graphene sheet giving rise to a magnetization along the out-of-plane i.e., +/- z direction! What is the mathematical expression of the incident circularly polarized field that is exciting the graphene disk? Also in which plane is the graphene disk situated?

A large part of this question is answered already in our response to question 6 of reviewer 1. The polarization actually does not play a significant role in this regard, i.e. the in-plane field would also be zero for linearly polarized radiation. This is due to the high sheet conductivity on resonance that leads to a partial screening within the disk at resonance, i.e. the external in plane field leads to the charge carriers motion until the external field is canceled out within the disk. The circularly polarized radiation was included by superposition of the solution with linearly polarized radiation of two perpendicular field directions and a phase shift of π .

(c) Do the theoretical simulations have the same periodicity (1.5 μm) as in the experiments? It is certainly not the case if the bounding box presented in figure 4 do represent the unit cell. Then why the periodicity is chosen to be different from the same in the experiment?

Yes, our simulation was carried out for a single graphene disk with periodic boundary condition with the

unit cell matching the real sample ($1.5 \mu\text{m} \times 1.5 \mu\text{m}$). We admit that Fig.4(b) can cause confusion about whether the simulation was carried out in the different periodicity. Thus, we changed Fig. 4(b) to show the entire unit cell.

(d) Is the periodic boundary condition in three dimension (3D) or is it a two-dimensional periodicity? If it is in 3D then did the authors check the convergence of the results with respect to the vacuum along the z direction in the simulation cell? If not, they should. The length of the arrows indicating the strength of the induced magnetic field do not show significant decrease while going away from $z=0$ plane suggesting that neighboring cells along z-directions has strong influence on the generated magnetic field.

The periodic boundary condition is a two-dimensional periodicity. In z-direction we implemented the source above the sample with an absorbing boundary within the SiC substrate (i.e. it behaves like a semi-infinite substrate). To avoid misunderstanding, we inserted the dimensional information in page 13 (220), as:

“... considering two-dimensional periodic boundary conditions”

(e) What does the unit cell (16 nm square) correspond to? The bounding box shown in the figure 4 has a dimension of approximately 600 nm x 1200 nm x 1200 nm.

We thank the referee for pointing out this issue, we meant actually the mesh size for the finite element calculation and not a unit cell in the sense of a periodic pattern. To avoid misunderstandings, we changed it in the manuscript to be clearer:

on the scale of the unit cell -> ...on the scale of the mesh size...

and the charge Q per unit cell -> and the charge Q per mesh

the area of the unit cell employed -> the mesh area employed

within a single unit cell -> within a single mesh

(f) It is not clear what do E_1 and E_0 correspond to! Please provide mathematical expressions to define them properly, or explain better.

We thank the reviewer for this comment, our explanation was certainly too short to be clear for the readers. The idea is to extract the local carrier density from the electric field. For this, we assume that there is only a minor change in carrier density on the scale of the mesh size, i.e. the electric field for each mesh stems approximately from a homogeneous charge carrier distribution within the disk. The factor α accounts for the difference between the electric field caused by the charges in a single unit cell vs. a large area at the

position of the disk with the identical charge carrier density in the single unit cell. This way, we can directly calculate the charge carrier density from the electric field above each mesh. To clarify this point to the reader, we added the following statement:

The additional factor $\alpha = 1.5 \cdot 10^4$ accounts for the ratio ($\alpha = E_1/E_0$) between the electric field E_1 generated from homogeneously distributed carriers and the field E_0 caused by the charges within a single unit cell: E_1 is in this case the electric field calculated via the finite element simulation. To derive the electric field caused by the charges within a single unit cell directly underneath the electric field, we calculate the ratio α between the electric field caused by the charges in a single unit cell and a homogeneous charge distribution with the same carrier density. By assuming that the charge carrier density is rather constant on the length scale of a unit cell, we can convert the electric field E_1 into the field E_0 , which represents the electric field caused by the charges in the unit cell.

(g) Is the current density \vec{J} is along the Cartesian y-direction as suggested by \hat{j} ? This would imply that the current is curl free and can not give rise to a magnetic field!

We thank the reviewer for pointing out this ambiguous statement. What was actually meant is that \hat{j} is the unit vector pointing in the azimuthal direction and thus leads to a current circulating the disk. We changed the symbol to $\hat{\varphi}$ as it is the more standard symbol for azimuthal angles and leads to less confusion. Furthermore, we changed the statement on page 14 (240) as follows:

“...in Cartesian coordinate. Note that the current density caused by the circular plasmons curls around the center of the disk. Hence, the unit vector of $\hat{\varphi}$ points into the azimuthal direction. For calculating...”

(h) Is conversion of the optical conductivity from 3D to 2D a standard one? If so, please give proper reference to support it, otherwise please give further justification.

The unit of bulk optical conductivity is $\Omega^{-1}\text{cm}^{-1}$ and the corresponding unit in two-dimension (2D) is Ω^{-1} . The standard conversion method from bulk to 2D and/or vice-versa is to consider the thickness of the sample. As for the proper references regarding this conversion, we added two references below to the manuscript:

1. L. Matthes et al., Optical properties of two-dimensional honeycomb crystals graphene, silicene, germanene, and tinene from first principles, *New J. Phys.* **16**, 105007 (2014).
2. Z. Zhou et al., Dimensional crossover in self-intercalated antiferromagnetic V_5S_8 nanoflakes, *Phys. Rev. B* **105**, 235433 (2022).

One can refer to the explicit conversion relation at Eq. (2) and (1) in the above references of 1 and 2,

respectively.

(i) Please provide details of the equations that are solved and of the numerical methods employed in the FEM code, at least in the supplementary material.

The electromagnetic fields were calculated in COMSOL using a finite element method solving Maxwell's equations. The simulation used periodic boundary conditions in the horizontal direction, to simulate the square lattice of graphene elements, and an absorbing boundary in the silicon carbide to mimic a semi-infinite substrate. We added this statement accordingly to the methods section.

Response to the comment of the referee 3

Introductory Comment:

Han et al present the use of a graphene microdisk array for the generation of high-intensity magnetic field pulses via plasmon-resonance enhanced inverted faraday effect without the need for the high pump intensities of previous reports. They deduce the field intensity produced via the faraday rotation of a probe beam and reveal the field-dynamics via polarization-resolved variable delay pump-probe spectroscopy. Further the work presents the advantageous of the microfabricated graphene arrays when compared to unpatterned graphene and accomplishes this using industry standard techniques (EBL, RIE) and substrates (SiC) which offer the potential for expansion of these concepts into many applications. With this noted, I believe that the work contains the novelty and broad impact to warrant publication in a journal such as nature communications.

There are, however, several issues which should be addressed before I can give my recommendation for publication.

We thank the referee for evaluating our work as important research providing new perspectives for many applications. Detailed point-by-point replies to each comment and changes to the manuscript are listed below.

1. Literature support needs to be improved in order to appeal to a broad audience. For instance

We tried to cite all publications relevant for the current manuscript. Apparently, the reviewer missed to list the ones relevant, so we once more went through earlier publications and found the following one that is added in the main manuscript on page: doi.org/10.1021/acs.nanolett.2c00571:

“...as it was recently shown for gold particles²⁴ and more recently for plasmonic arrays of gold nanodisks [Ref] with pump radiation in the visible range²⁴. The circularly...”

If we still missed relevant publications, we'd happily add those references.

1. Continuing from the previous point, the authors undersell the applicability of their arrays. They mention, “by placing molecules or nanoparticles in the vicinity of the disks” but do not elaborate (for instance) the potential for in-situ control allowed by optically induced magnetization.

We agree that the role of in-situ control via optically induced magnetization can be appealing for specific applications and added the statement as suggested on page 11:

“..... μm scale, enabling the potential applications for in-situ experiments. This allows....”

3. There is a significant overlap between Fig. 1a and Fig. 2. In truth it appears as though Fig. 1a is a subset of Fig. 2 with the pump-beam removed and does not add any extra information to the manuscript, and as the bolometers are used for the experiment presented in Fig. 1b I question if Fig. 1a's inclusion is needed. And to a lesser extent the Fig. 4 insert, especially as it is presented after the transient data of Fig. 3.

Figure 1a shows the experimental scheme for the reference measurement, i.e., static-magnetic field dependence of Faraday rotation. In contrast, the sketch of the experiment for the optically induced Faraday rotation was presented in Fig. 2. Thus, we believe that Fig. 1a is necessary. However, we agreed that future readers can run into the same confusion. To avoid this unnecessary confusion, we changed the title of figure caption 1, as:

“Faraday rotation θ_F of graphene disks and unpatterned graphene under static magnetic fields”

Fig. R3: Transmitted THz signal T_E as a function of the polarizer angle θ .

4. Can the authors comment on how the error bars in Fig. 1b were determined?

Response to comment 4: We collected transmitted THz signal T_E by rotating the angle of wire-grid polarizer θ . For the extraction of Faraday angle θ_F from the experiment data, we carried out fitting using Eq. (R1):

$$T_E = A \cdot \sin(\theta + \theta_F) + \text{Offset} \quad (\text{R1})$$

Figure R3 shows the experiment and the fitting result on the magnetic field of 6 T. From this fitting, one can obtain the standard error.

5. Can the authors comment on the relation between ω_p and the dimensions? In section S.1 the authors determine ω_p via FTIR spectroscopy, is there theoretical basis for the shift compared to unpatterned graphene?

First of all, we'd like to point out that propagating plasmons in unpatterned graphene do not have a fixed frequency, but can occur at any frequency. This should not be confused with the plasma frequency, which is determined by the scattering rate of the charge carriers. The propagating plasmons are difficult to excite optically and require patterning for excitation under normal incidence. In contrast to this, the localized plasmons in the structures reported in our manuscript essentially combine the plasmon dispersion with a spatially confined standing wave that can be directly excited optically under normal incidence. In this case, the structure size corresponds to half the wavelength of the standing wave, the frequency depends on the plasmon dispersion. A comprehensive explanation of the role of the disk dimension can be found in DOI:10.1021/nn3055835. As this might be of interest for other readers as well, we added the following statement on page 4:

“... plasmonic disks at a frequency of 3.5 THz, which depends on the dimensions of the disks [DOI:10.1021/nn3055835]. In this case the graphene disks have a diameter of 1.2 μm and are fabricated from quasi free...”

6. Fig. 1c&d present the calculated faraday rotation spectra for patterned/unpatterned graphene respectively. There is, however, no experimental data to confirm this. Is this because it is too experimentally tedious (e.g. would require a major reconfiguration of the FEL)?

The FEL itself is a high intensity, narrow bandwidth source that can be tuned in a large wavelength range. But as tuning over a larger frequency span requires lengthy realignment processes, a proper wavelength dependence is not possible, due to the lack of beam time available. For this type of linear Faraday rotation, a broadband source is better suited for such measurements, which have been carried out by other groups before (e.g. Ref. 33 in the main manuscript). We'd like to point out that the linear measurements of the Faraday rotation are only performed as reference measurement to compare it to the pump-induced Faraday rotation and not new by itself.

7. Regarding the last two points, were the array dimensions optimized to work with the FEL? If so, the authors should make this clear.

Yes, the samples are specifically tailored for measurements with the FEL. At this wavelength good quarter wave plates are available and the transmission in air is well in a large range. The density of the disks was chosen to achieve a strong light matter interaction without too much coupling between the disks (as described above). To make this point clear, we inserted the additional sentence on page 4:

“... electron-beam lithography. The dimensions of the disks and the array size were optimized for the experiments at FELBE. The periodicity of 1.5 μm leads to strong light-matter interaction and at the same time limits the coupling between the disks. To generate ...”

8. Why was 440 nJ cm^{-2} the upper limit of pump fluence? Is it a limitation of the experimental setup?

The answer to this question is twofold: the experimental setting at the time of the experiment did not allow a higher pump fluence, though it would have been technically possible by further optimizing the output power of the FEL. As stability of the FEL is more important for the experiment and also a significant saturation was already reached at that fluence, we did not strive to stronger pumping.

9. How exactly is θ_f determined (from $\Delta T/T$)?

The wire grid polarizer used to separate the two linear polarization components is set to 45° , i.e. both orthogonal contributions have the same power. The transmission through the polarizer as a function of the polarization angle is in general described by the Cos^2 between the electric field and the direction of the polarizer. When the polarization of the probe beam changes, the power for one bolometer increases following the Cos^2 -law, while the second one decreases accordingly. Thus, the difference between the relative changes of both channels has to be divided by two. Equation ER2 shows the equation exploited to

Fig. R4: Relation between $\Delta T/T$ and θ_f .

determine the Faraday angle. Figure R4 shows the relation between θ_F and $\Delta T/T$.

$$\text{Cos}^2(\theta_F+45^\circ)=\frac{1}{2}\left(\left.\frac{\Delta T}{T}\right|_{\sigma^+}-\left.\frac{\Delta T}{T}\right|_{\sigma^-}\right)+0.5 \quad (\text{ER2})$$

As these details might be interesting for other readers, we added the according section in the SI.

10. Following from the previous point. In Fig.3 (& several supp figures) $\Delta T/T$ and θ_f are not symmetrical for σ_{\pm} . Can the authors elaborate on this point? Pump ellipticity is discussed (~15%) however, the authors declare this effect to be negligible in the results. The authors should be careful to consider all other potential aspects of the experiment (e.g. Transmission dichroism, B1/B2 responsivities, disk fabrication astigmatism)

We agree that potential sources for the measured signals have to be excluded carefully in order to nail down the origin reliably. This is one aspect that lead us to the measurements with linearly polarized pump radiation. The polarization was picked to get the strongest effect on the polarization of the probe pulse, yet the pump-induced change in polarization was smaller than the change observed with circularly polarized pump radiation. As only a fraction of the pump radiation is actually linearly polarized, we can safely exclude the plasmonic effect reported previously as origin for the strong Faraday rotation. As we do not measure the static polarization, where windows or other optical effect might contribute, but differential, we would need to trigger another pump-induced Faraday effect, e.g. in the substrate. This we can also safely exclude as the measurements on the unpatterned graphene (all conditions maintained the same) did not show a significant change in polarization.

In conclusion, we are convinced that the effect is truly caused by the graphene disks. The difference between σ^+ and σ^- pumping is most likely due to the imperfect polarization state of the pump pulse as described above.

11. Following on from the previous point, would it be possible to tailor the ω_p based upon disk dimensions. How would fabrication astigmatism affect this (and the resultant inverted Faraday effect)?

The resonance plasmon frequency is mostly determined by the dimension, and slightly shifted by the coupling. Thus, as described above, it is possible to tailor the plasmon frequency in a large spectral range (see paper cited above). An anisotropic, i.e. elliptical, shape would lead to a dependence of the plasmon frequency on the polarization. But it could not cause a pump-induced Faraday rotation. Earlier test on other samples fabricated with the same e-beam tool did not reveal any anisotropy. To further exclude such effects, the polarization dependent transmission was measured and we could not find a difference regardless of the polarization. Thus, we can exclude that an elliptical shape of the disks would impact our measurements.

12. The material & device preparation methodology seems very similar to the author's previous work by

Woo et al. (Advanced Photonics Research 3.2 (2022): 2100218), This is peculiarly not referred to in the manuscript.

We noticed that this paper was unintentionally only cited in the supplementary. We added the reference accordingly on page 3 (63).

13. In section S6 polarization nonlinearity contribution to the signal is measured with a linearly polarized pump. The authors then use this as evidence that the contributions of the linear component of the pump to be negligible. This assumption hinges on another assumption that the circular polarized component also has negligible contribution to the plasmonic nonlinearity. This assumption is not supported by the experiment. A more complete picture would be achieved by including circular dichroism and/or faraday ellipticity to account for these contributions.

As explained above, the circularly polarized pump pulse only has a small linear contribution that could not explain the observed results. We respectfully disagree with the statement that the circular polarization would not lead to a plasmonic nonlinearity and we believe that this might be a little misunderstanding: the pump-induced circular dichroism as presented in (Advanced Photonics Research 3.2 (2022): 2100218) cannot be explained without such a plasmonic nonlinearity. It is rather the contrary: the observation in that earlier study triggered the current one. In nonconductive and nonmagnetic media, a pump-induced circular dichroism is not possible (see SI of earlier paper for details). This led us to the conclusion, that a significant magnetic field has to be caused by the circulating plasmons in order to break the symmetry, and therewith motivated our measurement of the pump-induced Faraday rotation. I.e. the effect that leads to the circular dichroism is the same one as the one that causes the Faraday rotation.

14. The Authors present experimental (and theoretical) results for Faraday rotation unpatterned graphene, however any results for the inverted faraday effect (I.e. Fig. 3) for unpatterned graphene are omitted. I believe this is because the off-diagonal optical conductivity (σ_{xy}) for unpatterned graphene around 3.5 THz to be negligible small. There is however, no rationale present in the text for its omission.

Indeed, the question of the role of the inverse Faraday effect in unpatterned graphene is an important point and was actually measured during the experiments. Figure R5 shows $\Delta T/T$ and θ_F obtained from the unpatterned graphene. As expected by the reviewer, the pump-induced Faraday rotation is much smaller in this case. To make this point clear to the readers, we added the following statement on page 7:

“...of the transmission. We experimentally confirmed that unpatterned graphene gives significantly weaker pump-induced Faraday rotation, emphasizing the role of the plasmonic structure. As discussed...”

Fig. R5: Experimental results of pump-induced transmission change $\Delta T/T$ and corresponding θ_F on unpatterned graphene.

5. What is the significance of the green filled trendline in figure 4? (Under the transient-snapshots)

The green-filled trendline was meant to express the passing time between the subfigures. It turned out that this is not quite understood in this way, so we omit this feature in the revised version.

16. Fig. S5 shows the temperature dependent induced faraday rotation, which the Authors use to determine the induced Magnetic field. Was the (simple) faraday rotation measured at each temperature prior to this measurement? If so, then this should be stated more clearly. If not, the Authors need to support why they expect the rotation measure at 10K is unchanged at elevated temperatures.

We thank the reviewer for pointing this out, actually we have measured the Faraday rotation in static field only at low temperature. Thus, the assignment of a corresponding magnetic field is not helpful in this regard, as we cannot assure that it does not change at elevated temperature. We have avoided the magnetic field axis in the revised version of Fig. S5.

17. The authors should consider adding some form of microscopy of the arrays (either optical or SEM) to both enhance the reader understand the structure and to help qualitative evaluation of the structures.

Unlike graphene on thin optical films, epitaxial graphene that is grown on silicon carbide is rather difficult to observe with an optical microscope, SEM is known to damage graphene, so we performed AFM measurements on related samples fabricated in an identical process (cf. *Advanced Photonics Research* 3.2 (2022): 2100218). As described above, in that earlier study we also present Raman measurements on the graphene.

Reviewers' Comments:

Reviewer #1:

Remarks to the Author:

I am satisfied that the authors have adequately addressed all of my concerns, as well as the points raised by the other referees. I believe the manuscript is ready for publication.

Reviewer #2:

Remarks to the Author:

Please find below the comments and questions to the reply of the authors.

1. Regarding reply to question 1a and 1b.

What is CST simulation? Is this different from the COMSOL simulations? Please give details or references to what quantity is shown and how they are calculated in Fig. R1. I am sorry that I'm ignorant about CST simulation and that I need more information. At the same time, I think that as Nat. Comm. covers wide range of audience it is customary to explain in details, either in the main text or in the supplementary material.

2. Regarding reply to question 4a.

(i) What are the artifacts the authors mean in their reply when they say "to avoid artifact stemming from the charge distribution within the disk"?

(ii) The answer to the question (Why the electric field is calculated 50 nm above the graphene disk?) is unfortunately not obtained! Why is it 50nm and not 5nm or 500 nm?

(iii) The authors wrote that "The calculated maximum magnetic field is about 0.35 T, which is on the same order of magnitude as the corresponding experimental value (0.7 T)." Now, if I understood correctly, they calculate electric fields (say, E) in COMSOL, and then use eq.(3) to calculate charge density (Q) [which depends on ξ (chosen to be 50nm)] which give Δn which gives the J responsible for the magnetic field. However, they do not mention how much this value of B (0.35 T) would change if they would calculate the E at 5nm or 500 nm. They should either show this dependence to support their statement that they found calculated values to be of the same order of magnitude to the experimental one, or they should explicitly mention that the fact that they found the calculated value to be of the same order of magnitude to the experimental one belongs to the choice that the fields are calculated at 50nm above the graphene disk. Otherwise in my humble opinion it might be prone to a bit of overstatement!

3. Regarding reply to question 4b.

I am sorry, I do not understand the reply of the authors to why the in-pane component E_x and E_y are neglected! In the supplementary information equation (S10) it is clear that if $E_x=0$ and $E_y=0$ then there will be no J_x and J_y . Nevertheless, in the corrected equation (4) the authors shows that the current \vec{J} is along the $\hat{\phi}$.

(i) The authors wrote "In our calculation, we used Cartesian coordinate and ϕ is the unit vector of the direction of current density represented in Cartesian coordinate." and they also wrote "the unit vector of ϕ points into the azimuthal direction". Is $\hat{\phi}$ a unit vector in the X-Y (or a different $z=\text{constant}$) plane or not?

(ii) If yes, then J_x and J_y are not zero. And then, following equation (S10) E_x and E_y can't be zero. Then the authors need to explain why the in-pane component E_x and E_y are neglected.

(iii) The derivation of optical conductivity tensor explained in Section S2 is not clear. For example, the authors don't mention the directions of fields. Please, give a clearer derivation.

(iv) What is the meaning of "partial screening" here? What/who is screening what/who from what/who? What the authors wanted to express by "... the external in plane field leads to the charge carriers motion until the external field is canceled out within the disk" is not clear to me at all. Could they refer to equations, or mathematical expressions that would better describe their statement?

Apart from these, I did not get my answers to the following questions:

- (iv) What is the mathematical expression of the incident circularly polarized field that is exciting the graphene disk in the simulations?
- (v) Also in which plane is the graphene disk situated in the simulations?

4. Regarding reply to question 4e.

- (i) Are the COMSOL simulations done in 3d or in 2d? From the fig 4b it seems that the B field is calculated in three dimension. Then I do not understand why the mesh is two-dimensional!
- (ii) Are the authors calculating the charge density at a surface? If yes, what is this surface?

5. Regarding reply to question 4f.

The yellow highlighted text in the reply document is not the same as the one in the main article! In the reply document what followed the phrase "...To clarify this point to the reader, we added the following statement:" contains 'unit cell' instead of 'mesh'! So, it takes quite a bit of time to go back and forth between the two documents to understand that what the authors wanted to mean by "By assuming that the charge carrier density is rather constant on the length scale of a unit cell" in the reply document is that density is rather constant on the length scale of a mesh! These mistakes which are simple ones give the impression that the work is done with less amount of care than it required. Nevertheless, even after understanding that the authors forgot to replace 'unit cell' by 'mesh' in the reply document, it is not clear for me what does E1 and E0 correspond to.

- (i) "E1 generated from homogeneously distributed carriers". Where are these carriers coming from? Are the carriers homogeneously distributed over the whole extent of the unit cell?
- (ii) "E1 is in this case the electric field calculated via the finite element simulation." Calculated for which source of charge? Is there an external electric field in the simulation or not? If so, what is the relation of the external field to E1?
- (iii) I suggest the authors to define explicitly what is the external field, what is the induced field, what source of charges giving rise to total field, etc., and be consistent with the naming of the variables throughout the article and the supplementary material.

6. Regarding reply to question 4i.

Although the authors replied that they solve Maxwell's equations in COMSOL, in my humble opinion it is not sufficient to understand what did they actually simulate, especially because they did not provide an answer to the question regarding the explicit expression of the incident circularly polarized external field!

- (i) Are the COMSOL simulations standard ones? Can the authors guide to some references (article/reviews, etc.) where we can understand what exactly the authors wanted to simulate?
- (ii) If not, in my humble opinion the authors need to give details of the equations (even if they are known Maxwell's equation) in relation to the problem they are solving. Why not write a section in the supplementary material to give the theoretical details?
- (iii) Probably, they can also present a schematic diagram of the boundary-value problem they are solving in COMSOL? This would help, e.g., to understand if they are solving for TE or TM modes.

General comments:

I think the manuscript presents important experimental results on the generation of plasmon-induced magnetization in graphene disks. However, the theoretical support to the experimental results lessens the experimental findings which is the principal message. Noticing the lack of theoretical details and explicit information on the simulations it felt like the role of the simulation was more to find a value comparable to experimental results than understanding the underlying mechanism. While in experiments we don't have control over all the parameters that influence the results in modeling and simulation we do. Which makes it possible to play with different parameters of the model and come up with results that we want. For this very reason, it is important to give every detail of the modeling and the simulations so that (1) readers can understand what choice of modeling parameters give results close to the experiments, and (2) the calculated results can be reproduced.

Conclusion:

Although the experimental results in the manuscript should be reported I do not consider the manuscript ready for publication with the present insufficient details and not completely comprehensible explanation on the theory and simulation presented in the manuscript and the supplementary material.

Reviewer #3:

Remarks to the Author:

With the revisions to the manuscript by han et al. the Authors were able to adequately address all of my concerns regarding their work.

Any remaining concerns previous raised (such as maximum pump fluence) would require extensive experimental efforts whilst providing only minimal boon.

It is thus my conclusion that the work presented is of sufficient novelty and scientific rigor to merit publication in Nature Communications.

Response to the reviewers

We are grateful for the reviewers careful reading of the manuscript and the helpful comments provided. In particular, we are thrilled about the positive feedback from reviewers one and three and address the comments of reviewer two below.

In the following we address (black) the reviewers' comments and questions (blue). Changes in the manuscript are highlighted in yellow.

Referee 2

Introductory Comment:

I think the manuscript presents important experimental results on the generation of plasmon-induced magnetization in graphene disks. However, the theoretical support to the experimental results lessen the experimental findings which is the principal message. Noticing the lack of theoretical details and explicit information on the simulations it felt like the role of the simulation was more to find a value comparable to experimental results than understanding the underlying mechanism. While in experiments we don't have control over all the parameters that influence the results in modeling and simulation we do. Which makes it possible to play with different parameters of the model and come up with results that we want. For this very reasons, it is important to give every details of the modeling and the simulations so that (i) readers can understand what choice of modeling parameters give results close to the experiments, and (2) the calculated results can be reproduced.

We thank the reviewer for the overall positive feedback and agree that every publication should give all information necessary to reproduce the results.

1. Regarding reply to question 1a and 1b.

What is CST simulation? Is this different from the COMSOL simulations? Please give details or references to what quantity is shown and how they are calculated in Fig. R1. I am sorry that I'm ignorant about CST simulation and that I need more information. At the same time, I think that as Nat. Comm. covers wide range of audience it is customary to explain in details, either in the main text or in the supplementary material.

CST Studio is, like COMSOL, a commercial software to solve Maxwell's equations via finite element methods. While both tools share the main functionality and capabilities, they are quite different in terms of the user interface. We have exclusively employed COMSOL for the calculations presented in the manuscript, and so this does not present any inconsistency for the readers. We employed CST in only our response to

the first round of reviews, in order to address a question about the coupling between neighboring disks. While COMSOL could have been used for these supporting simulations, CST greatly facilitates the computation of far-field plane-wave reflection and transmission from periodic surfaces, whereas COMSOL provides more convenient access to the electromagnetic field values within the computation window. We'd like to emphasize that this is not a fundamental limitation: we work with both software products and confirm that they give consistent results.

As the CST results were only used to demonstrate the role of the coupling of the disks and not part of the main manuscript, this information is only available in the response letter and we don't intend to add it to the main text.

2. Regarding reply to question 4a.

(i) What are the artifacts the authors mean in their reply when they say "to avoid artifact stemming from the charge distribution within the disk"?

In the simulations, we model the two-dimensional graphene as a thin ($d = 50$ nm) material with a bulk conductivity $\sigma_{3D} = \sigma_{2D}/d$. This is a commonly-used approximation that allows for commercial 3D simulation tools accurately simulate 2D boundaries. From Maxwell's equations, the areal sheet charge density at an interface is proportional to the normal component electric field at the surface. When the 2D material is modeled by a thin conductive dielectric, the charge density within the film is instead distributed not on the surface but throughout the thin film. Under these conditions, the normal electric field at a small distance above (but not directly on) the interface provides a better estimate of the equivalent cumulative areal charge density in the film, without compromising the spatial resolution.

(ii) The answer to the question (Why the electric field is calculated 50 nm above the graphene disk?) is unfortunately not obtained! Why is it 50nm and not 5nm or 500 nm?

As mentioned in our answer to (i), a direct readout at the surface would not be ideal, but at the same time we want to make sure to be close enough to the surface to get a reliable result. To evaluate the role of the distance, we actually performed the calculations for three different distances: 50 nm, 100 nm, and 200 nm. The calculation based on the field in 100 nm distance led to a moderate decrease of the resulting field of 15%. In contrast, a distance of 200 nm resulted in a field that was decreased by about 45%. From these values we can derive a decay length for an exponential decay that corresponds to about 180 nm. Assuming that the field would further follow this exponential decay, we can extrapolate that we underestimate the field by not more than 20%. Thus, we believe that the readout at 50 nm is a good compromise with reasonable errors that are far below an order of magnitude.

(iii) The authors wrote that "The calculated maximum magnetic field is about 0.35 T, which is on the same order of magnitude as the corresponding experimental value (0.7 T)." Now, if I understood correctly, they calculate electric fields (say, E) in COMSOL, and then use eq.(3) to calculate charge density (Q) [which depends on ξ (chosen to be 50nm)] which give Δn which gives the J responsible for the magnetic field. However, they do not mention how much this value of B (0.35 T) would change if they would calculate the E at 5nm or 500 nm. They should either show this dependence to support their statement that they found calculated values to be of the same order of magnitude to the experimental one, or they should explicitly mention that the fact that they found the calculated value to be of the same order of magnitude to the experimental one belongs to the choice that the fields are calculated at 50nm above the graphene disk. Otherwise in my humble opinion it might be prone to a bit of overstatement!

We agree that cherry picking the distance to get a matching result would not be proper science and shall never be done! As described above, we performed additional calculations to evaluate the role of the distance and to make sure that 50 nm is giving a reliable, trustworthy value.

3. Regarding reply to question 4b.

I am sorry, I do not understand the reply of the authors to why the in-plane component E_x and E_y are neglected! In the supplementary information equation (S10) it is clear that if $E_x=0$ and $E_y=0$ then there will be no J_x and J_y . Nevertheless, in the corrected equation (4) the authors shows that the current \vec{J} is along the $\hat{\phi}$.

The sheet charge carrier density and the sheet current density are of course related to one another by the continuity equation $\nabla \cdot J = -\frac{\partial \rho_V}{\partial t}$. This means that a time-varying charge density will necessarily produce a corresponding current density J , and one need not calculate both. One could, in principle, calculate J_x and J_y from the in-plane components of the E-field, but because the surface is highly conductive, the in-plane fields at the surface will be small. By contrast, the normal component of the electric field is sizeable, easily computed from simulations, and directly provides a measure for the scalar surface charge density.

(i) The authors wrote "In our calculation, we used Cartesian coordinate and ϕ is the unit vector of the direction of current density represented in Cartesian coordinate." and they also wrote "the unit vector of ϕ points into the azimuthal direction". Is $\hat{\phi}$ a unit vector in the X-Y (or a different $z=\text{constant}$) plane or not?

While we tried to rephrase the equation in the easiest possible way to give the reader an intuitive understanding of the calculation, we came to realize that the introduction of ϕ ("borrowed" from polar coordinates) did not help at all. Thus, we decided to describe everything in Cartesian coordinates in a clearer way, changing the $\hat{\phi}$ to \hat{r}_\perp as follows:

Note that the current density caused by the circular plasmons curls around the center of the disk. Hence, the unit vector of $\hat{r}_\perp = \frac{(\pm y, \mp x)}{\sqrt{x^2+y^2}}$ points into the azimuthal direction with the upper sign for clockwise rotation and the lower sign for counter-clockwise rotation.

(ii) If yes, then J_x and J_y are not zero. And then, following equation (S10) E_x and E_y can't be zero. Then the authors need to explain why the in-plane component E_x and E_y are neglected.

As described above, the in-plane fields are small due to the high conductivity of the graphene disk and actually do not give direct access to the carrier density. The out-of-plane component is much stronger and it is straight forward to calculate the charge carrier distribution this way.

(iii) The derivation of optical conductivity tensor explained in Section S2 is not clear. For example, the authors don't mention the directions of fields. Please, give a clearer derivation.

The optical conductivity tensor is a two-dimensional one due to the two-dimensional nature of graphene. Only in-plane electrical field components drive a current and are considered. Again we would like to emphasize that the external in-plane field is not zero and thus drives a current. The naming of σ_{xx} refers to the current generated in the direction of an in-plane electric field, while σ_{xy} refers to a current perpendicular to the driving field, which is caused in our case by the Lorentz force due to a magnetic field.

(iv) What is the meaning of "partial screening" here? What/who is screening what/who from what/who? What the authors wanted to express by "... the external in plane field leads to the charge carriers motion until the external field is canceled out within the disk" is not clear to me at all. Could they refer to equations, or mathematical expressions that would better describe their statement?

The term "screening" is well established in textbooks about electromagnetism and refers to the effect described above: the external field drives a current, the charge carriers thus generate an internal field that counteracts the external field, which is then (partially) canceled out (depending on the imaginary part of the conductivity). We believe that the additional explanation above is sufficient to explain the situation to the readers.

Apart from these, I did not get my answers to the following questions:

(iv) What is the mathematical expression of the incident circularly polarized field that is exciting the graphene disk in the simulations?

The electric field is given as a superposition of two linearly polarized components with a phase shift of 90° , the field pointing in plane with the graphene disk (as discussed above, an out of plane component would

not interact with the electrons in the disks):

$$E(t) = E_0 (\sin(\omega t)\hat{E}_x \pm \cos(\omega t)\hat{E}_y),$$

with E_0 being the amplitude of the field, the sign of the cosine term determines whether the polarization is clockwise or counterclockwise. We added the information as follows on Page 13:

“The value of the incident plane wave $E_{\text{THz}}(t)$ was set to a value of $E_{\text{THz}}^0 = 5.75 \cdot 10^5 \text{ V m}^{-1}$, which is equivalent to the highest field strength in the pump-probe experiments:

$$E_{\text{THz}}(t) = E_{\text{THz}}^0 (\sin(\omega_{\text{simul}} t)\hat{E}_x \pm \cos(\omega_{\text{simul}} t)\hat{E}_y) \quad (3)$$

where the sign of the cosine term determines the rotation direction of the polarization.”

(v) Also in which plane is the graphene disk situated in the simulations?

As mentioned in the previous answer, the disks are in the x-y plane. To make the whole geometry more accessible to the reader, we added a sketch in the SI.

Fig. R2_2: The geometry for COMSOL simulation.

4. Regarding reply to question 4e.

(i) Are the COMSOL simulations done in 3d or in 2d? From the fig 4b it seems that the B field is calculated in three dimension. Then I do not understand why the mesh is two-dimensional!

The COMSOL simulations are three dimensional, though it doesn't make much sense to read out the three-dimensional field distribution. Thus, we read out the electrical field in a plane 50 nm above the graphene disk.

(ii) Are the authors calculating the charge density at a surface? If yes, what is this surface?

As stated above, from Gauss' law, the areal sheet charge density at an interface is proportional to the normal component electric field at the surface. When the 2D material is modeled by a thin conductive dielectric (as is done here), the charge density within the film is instead distributed not on the surface but throughout the thin film. Under these conditions, the normal electric field at a small distance above (but not directly on) the interface provides a good estimate of the equivalent areal charge density in the film.

5. Regarding reply to question 4f.

The yellow highlighted text in the reply document is not the same as the one in the main article! In the reply document what followed the phrase "...To clarify this point to the reader, we added the following statement:" contains 'unit cell' instead of 'mesh'! So, it takes quite a bit of time to go back and forth between the two documents to understand that what the authors wanted to mean by "By assuming that the charge carrier density is rather constant on the length scale of a unit cell" in the reply document is that density is rather constant on the length scale of a mesh! These mistakes which are simple ones give the impression that the work is done with less amount of care than it required. Nevertheless, even after understanding that the authors forgot to replace 'unit cell' by 'mesh' in the reply document, it is not clear for me what does E1 and E0 correspond to.

We would like to apologize, this was a mishap exclusively in the response letter that happened due to the subsequent answering of the reviewer questions and was overlooked in the response letter, in the main manuscript the naming is correct! Regarding the question about E1 and E0: E1 is the field calculated via COMSOL and caused by a charge carrier distribution that can be considered homogeneous in a large area (large compared to the mesh size and the distance to the readout plane 50 nm above the graphene sheet). To infer the local charge carrier density in a specific mesh from the electric field directly above it, we have to consider the ratio between the electric field that is caused only by the local charges E0, and the electric field calculated by COMSOL.

(i) "E1 generated from homogeneously distributed carriers". Where are these carriers coming from? Are the carriers homogeneously distributed over the whole extent of the unit cell?

The effect of a homogeneous carrier distribution is easy to calculate, what is meant by this is an infinite plane with constant carrier density. Obviously, in the calculation the carrier distribution is not homogeneous on an infinite scale, but varies significantly within the disk. Nevertheless, the impact of the variations on the local field are small: When the lateral distance to a specific point above the disk is significantly larger than the distance between the read-out plane and the disk, the impact is negligible. This is essentially equivalent to the assumption that we have perpendicular fields. For larger distances, a calculation of the carrier density might be possible, but much more difficult. The number of carriers within the disk is conserved: the ones that are added on one side are missing on the other.

(ii) "E1 is in this case the electric field calculated via the finite element simulation." Calculated for which source of charge? Is there an external electric field in the simulation or not? If so, what is the relation of the external field to E1?

The external field is a pure in-plane field (plane wave), so it has not out of plane component! Thus, the perpendicular fields are solely caused by the carrier distribution within the disk, which is driven by the external in-plane fields.

(iii) I suggest the authors to define explicitly what is the external field, what is the induced field, what source of charges giving rise to total field, etc., and be consistent with the naming of the variables throughout the article and the supplementary material.

We revised the manuscript and made sure to use consistent terminology, i.e. external field for the incoming radiation, internal field for the fields stemming from the electrons in the disks, and net field for the combined field. The changes are marked in yellow in the manuscript.

6. Regarding reply to question 4i.

Although the authors replied that they solve Maxwell's equations in COMSOL, in my humble opinion it is not sufficient to understand what did they actually simulate, especially because they did not provide an answer to the question regarding the explicit expression of the incident circularly polarized external field!

(i) Are the COMSOL simulations standard ones? Can the authors guide to some references (article/reviews, etc.) where we can understand what exactly the authors wanted to simulate?

Yes, COMSOL is a standard tool to calculate field distributions for exactly such situations and commonly used for plasmonics (e.g. 10.1103/PhysRevX.5.031029, 10.1063/5.0153032, 10.1016/j.procs.2017.12.049, and many more). The second part of the question is a bit unclear, as we clearly describe that we used

COMSOL to calculate the electric field distribution in the vicinity of the graphene disk. While a comprehensive explanation of finite element electromagnetics is clearly beyond the scope of the manuscript, we believe that, with the expressions clearly given for the circularly polarized incident electric field, and with the additional references provided, our revised manuscript gives sufficient information to permit any interested reader to both understand the calculations performed and to repeat them if necessary.

(ii) If not, in my humble opinion the authors need to give details of the equations (even if they are know Maxwell's equation) in relation to the problem they are solving. Why not write a section in the supplementary material to give the theoretical details?

As mentioned above, the details of the software are far beyond the scope of the current study as we use it as an existing tool. To give the readers who might be interested in the details of finite element calculations additional information, we added the following citation as reference for comprehensive knowledge about the finite-element method:

Hutton, D. V. Fundamentals of finite element analysis. McGraw-Hill Science: New York, NY, USA (2004).

(iii) Probably, they can also present a schematic diagram of the boundary-value problem they are solving in COMSOL? This would help, e.g., to understand if they are solving for TE or TM modes.

We believe that the geometry and the polarization is more clear to the reader as we added the sketch mentioned above.

Additional remark: the author contribution of Wojciech Knap has been missing and is included now.

Reviewers' Comments:

Reviewer #2:

Remarks to the Author:

The authors have taken into account the concerns regarding the theoretical support to the experimental results and responded to my satisfaction. I consider the manuscript to be ready for publication. While the publication remains subject to the decision of the editor I take the opportunity to congratulate the authors for their work.

Response to the reviewers

We are grateful for the reviewers careful reading of the manuscript and the valuable comments to improve our manuscript! Below is the final reviewer comment as well as the final changes to the manuscript.

Referee 2

Introductory Comment:

The authors have taken into account the concerns regarding the theoretical support to the experimental results and responded to my satisfaction. I consider the manuscript to be ready for publication. While the publication remains subject to the decision of the editor I take the opportunity to congratulate the authors for their work.

Final corrections/modifications to the manuscript include the introduction of section (marked in yellow) and subsection headlines. In addition, we reorganizing the order of sections according to the editorial requirements of Nature Communications, including the fonts for equations.